# Ciliary Neurotrophic Factor (CNTF) and Its Receptors Signal Regulate Cementoblasts Apoptosis through a Mechanism of ERK1/2 and Caspases Signaling

**DOI:** 10.3390/ijms23158335

**Published:** 2022-07-28

**Authors:** Jiawen Yong, Sabine Groeger, Julia von Bremen, Sabine Ruf

**Affiliations:** 1Department of Orthodontics, Faculty of Medicine, Justus Liebig University Giessen, D-35392 Giessen, Germany; sabine.e.groeger@dentist.med.uni-giessen.de (S.G.); julia.v.bremen@dentist.med.uni-giessen.de (J.v.B.); sabine.ruf@dentist.med.uni-giessen.de (S.R.); 2Department of Periodontology, Faculty of Medicine, Justus Liebig University Giessen, D-35392 Giessen, Germany

**Keywords:** cementoblasts, CNTF, apoptosis, ERK1/2, caspase, CNTF-receptors

## Abstract

Ciliary neurotrophic factor (CNTF) was identified as a survival factor in various types of peripheral and central neurons, glia and non-neural cells. At present, there is no available data on the expression and localization of CNTF-receptors in cementoblasts as well as on the role of exogenous CNTF on this cell line. The purpose of this study was to determine if cementoblasts express CNTF-receptors and analyze the mechanism of its apoptotic regulation effects on cementoblasts. OCCM-30 cementoblasts were cultivated and stimulated kinetically using CNTF protein (NBP2-35168, Novus Biologicals). Quantified transcriptional (RT-qPCR) and translational (WB) products of CNTFRα, IL-6Rα (CD126), LIFR, p-GP130, GP130, p-ERK1/2, ERK1/2, Caspase-8, -9, -3 and cleaved-caspase-3 were evaluated. Immunofluorescence (IF) staining was applied to visualize the localization of the CNTF-receptors within cells. The apoptosis ratio was measured with an Annexin-V FITC/PI kit. The ERK1/2 antagonist (FR180204, Calbiochem) was added for further investigation by flow cytometry analysis. The CNTF-receptor complex (CNTFRα, LIFR, GP130) was functionally up-regulated in cementoblasts while cultivated with exogenous CNTF. CNTF significantly attenuated cell viability and proliferation for long-term stimulation. Flow cytometry analysis shows that CNTF enhanced the apoptosis after prolonged duration. However, after only a short-term period, CNTF halts the apoptosis of cementoblasts. Further studies revealed that CNTF activated phosphorylated GP130 and the anti-apoptotic molecule ERK1/2 signaling to participate in the regulation of the apoptosis ratio of cementoblasts. In conclusion, CNTF elicited the cellular functions through a notable induction of its receptor complex in cementoblasts. CNTF has an inhibitory effect on the cementoblast homeostasis. These data also elucidate a cellular mechanism for an exogenous CNTF-triggered apoptosis regulation in a mechanism of ERK1/2 and caspase signaling and provides insight into the complex cellular responses induced by CNTF in cementoblasts.

## 1. Introduction

The ciliary neurotrophic factor (CNTF) was initially discovered because of its capability to support the survival of parasympathetic neurons of the chick ciliary ganglion in vitro [1] and subsequently purified to homogeneity from sciatic nerves [2]. It belongs to the family of interleukin (IL)-6 related cytokines [3] including leukemia inhibitory factor (LIF), IL-6, IL-11 and IL-17 [2].

CNTF exerts its biological functions by binding with high affinity to a receptor complex consisting of the CNTF α-receptor (CNTFRα), the β-receptor glycoprotein 130 (GP130) and leukemia inhibitor factor receptor (LIFR). Schuster et al. (2003) showed that the IL-6 receptor (IL-6R) can serve as an α-receptor for CNTF [4], and CNTF elicits cellular responses with low affinity to the tripartite complex of CNTRα, IL-6R and LIFR. Importantly, CNTF and IL-6 type cytokines are elicited by different multiunit receptor complexes that always include the membrane-spanning 130-kDa glycoprotein, GP130 [5]. CNTF first binds in a 1:1 stoichiometry to the CNTFRα, which is not involved in the signaling transduction [6]. Binding of CNTF to the membrane-bound or soluble CNTFRα induces a heterodimer of the signal transducing β-receptor GP130 and LIFR, which triggers intracellular signaling cascades [4]. The formation of the CNTF-receptor complex triggers the activation of multiple intracellular signaling pathways such as Ras/MAPK [7], which mediate cell survival, proliferation and apoptosis. It is reported that CNTF and its receptors are expressed in various tissues such as the brain, spinal cord and ciliary ganglia [8], as well as in non-neuronal cells such as human multipotent adipose-derived stem cells [9], myocytes [10], prostatic cells [11], osteoblast-like cells [5] and pancreatic β-cells [12].

Cementoblasts, highly differentiated periodontal ligament (PDL) cells, are located on the cementum [13]. The apoptotic death of cementoblasts might contribute to orthodontically induced inflammatory root resorptions (OIIRR) during orthodontic tooth movement (OTM) [14]. One of the most distinctly activated apoptotic signals was the caspase (cysteinyl aspartate proteinases) family, which executes the extrinsic and intrinsic apoptosis pathways [15]. Caspase-9 functions to activate effector caspases (Caspases-3, -6 and -7) and are involved in the execution of apoptosis. Caspase-8, an initiator caspase, activates downstream effector caspases. Caspase-3, the primary executioner of apoptotic death, results in apoptosis by cleaving numerous cellular proteins [15,16]. To the best of our knowledge, little is known about the role of CNTF and its receptors in cementoblasts. The mechanism of apoptotic regulation by CNTF also remains unclear. Our research aimed to clarify the regulatory effects of CNTF on the apoptosis of cementoblasts and the possible interaction with its corresponding signaling pathways.

## 2. Results

### 2.1. Ciliary Neurotrophic Factor Facilities the Expression of CNTF-Receptor Complex mRNA and Protein in Cementoblasts

In order to investigate whether CNTF signals through the receptor complex which includes CNTFRα, IL-6Rα (CD126), LIFR and GP130, the transcriptional and translated products of these receptors were compared in CNTF kinetic stimulated groups.

WB results demonstrated that the protein expression of CNTFRα, LIFR and IL-6Rα were induced by CNTF in cementoblasts (Figure 1A,B). The CNTFRα was sustained expressed over a period of 48 h. The LIFR was activated 1 h after CNTF addition, reaching a peak at time point 48 h. The IL-6Rα was expressed in cementoblasts, being the WB’s bands detectable during 5 to 90 min (Figure 1A). Despite our attempts, the IL-6Rα was unable to be detected for a longer period (2 to 48 h) with stimulation by CNTF protein in cementoblasts (Figure 1B).

The mRNA expression of *CNTFRα*, *LIFR* and *IL-6Rα* of cementoblasts are represented as the RT-qPCR results (Figure 1C). There were significant increases in mRNA levels of *CNTFRα*, *LIFR*, *IL-6Rα* after CNTF addition. Therefore, cementoblasts express CNTF-receptor complex at mRNA and protein levels.

Complementary to the WB and RT-qPCR results, the IF staining localized CNTFRα, LIFR and IL-6Rα proteins in cementoblasts in vitro. The representative picture of each receptor component is shown in Figure 1D. The green signal for CNTFRα is located in the cytoplasm, perinuclear region and cytomembrane (red arrow). The immunopositivity of LIFR is located mainly nuclear, while the cytoplasm is moderately stained (yellow arrow). IL-6Rα were weakly detected mainly in the cytoplasm and membrane. The staining pattern for all three receptor components appeared punctate, resembling typical membrane proteins.

### 2.2. CNTF Induces Initial Activation of the Cytokine Receptor GP130 and Promotes GP130 Phosphorylation in OCCM-30 Cells

We further study whether CNTF can induce GP130 in cementoblasts. WB results demonstrated that the expression level of GP130 was detectable during 5 to 15 min of CNTF stimulation. After 45 min, GP130 protein was only weakly or not detectable during WB analysis (Figure 2A).

In a similar manner, CNTF induced GP130 phosphorylation was significantly up-regulated during 5 to 15 min and the GP130 was sustained phosphorylated during 1 to 24 h (Figure 2B). Accordingly, the quantitative mRNA analysis showed that *GP130* was increased significantly after CNTF addition (*p* < 0.005).

In order to characterize the cellular localization of GP130 and p-GP130, we perform the IF study in CNTF treated cementoblasts. Both GP130 and p-GP130 protein appeared in the cytoplasm and cytomembrane, resembling typical membranal and cytoplasmic proteins (Figure 2D,E).

### 2.3. CNTF Decreases the Cell Viability, Proliferation and Induced Phosphorylated ERK1/2 Expression of Cementoblasts

Next, we analyzed the effect exogenous CNTF stimulation exerts on cementoblasts homeostasis. Cell viability and proliferation were evaluated by MTS assay and the IF staining of cellular proliferation marker Ki-67. The IF staining showed that the proliferation activity of cementoblasts was significantly inhibited with CNTF stimulation by 2.4-fold compared to the control group (Figure 3A,B). MTS cell viability assay was performed to evaluate the OCCM-30 cells survival in response to CNTF and IL-6, the latter of which acted as a positive control. The cell viability was significantly decreased (0.9-fold, *p* < 0.05) after CNTF addition at 400 ng/mL (Figure 3C), and concentration was determined based on a dose-kinetic analysis of CNTF on cementoblast viability. These results revealed that treatment of CNTF reduced the cell viability and proliferation of cementoblasts.

Furthermore, CNTF induced the phosphorylation of ERK1/2 expression during 5 to 30 min in a time-dependent manner (Figure 3D,E). These results suggested that CNTF may mediate the homeostasis of cementoblasts by activation of ERK1/2 MAPK signaling.

### 2.4. CNTF Regulates Cementoblasts Apoptosis and Activates the Caspases Signaling Pathway

To elucidate the influence of exogenous CNTF on the apoptosis ratio of cementoblasts, cells were cultured with CNTF for indicated time periods. Flow cytometric analysis showed that the apoptosis rate was significantly decreased within 1 h (5.12% ± 0.42%) in comparation with the control group (7.33% ± 0.34%) (Figure 4A,B). Interestingly, it was significantly elevated after 24 h co-stimulation of CNTF (16.81% ± 1.98%) compared with the control group (7.33% ± 0.34%) (Figure 4A,B). Meanwhile, the protein expression of Caspase-8 and -9 were not detected during 5 to 15 min after CNTF addition (Figure 4C).

On the contrary, when co-treated with CNTF, the Caspase-8 and -3 proteins as well as their cleaved forms were significantly increased during 15 min to 2 h (Figure 4C). After long-term treatment, the Caspase-8, -9 and -3 were significantly increased by CNTF (Figure 4D). Furthermore, the mRNA expression of *Caspase-8*, *-9* and *-3* were significantly up-regulated at 24 and 32 h (Figure 4E). These results suggest that CNTF protects cementoblasts from apoptosis within 1 h, whereas it promotes its apoptosis at long-term exposure.

### 2.5. ERK1/2 Is Involved in the Modulation of CNTF-Induced Apoptosis

In the last step, the apoptotic regulation of ERK1/2 exerted on CNTF-induced apoptosis was investigated by an ERK1/2 antagonist (FR180204). As shown by flow cytometric analysis, on selective suppression of ERK1/2, a significant increase in the cells apoptotic rate (17.50% ± 3.20%) (Figure 5A,B) can be seen. While undergoing the co-stimulation of CNTF and FR180204, the up-regulated apoptotic rate was significantly reversed by the addition of CNTF (10.81% ± 1.33%) within 1 h, suggesting CNTF has a protective effect on the apoptosis of cementoblasts at short-term periods (Figure 5A,B). These findings confirmed that ERK1/2 blockade increases cementoblasts apoptosis ratio and that CNTF has a varying level of protective effect regarding apoptosis progress within short-term exposure periods.

## 3. Discussion

This is the first study to provide direct evidence that cultured cementoblasts constitutively express CNTF tripartite receptor complex including CNTFRα, GP130, LIFR and IL-6Rα as well as their localization. Furthermore, phosphorylation of GP130 was induced by CNTF as a signal transducer. CNTF further activates the ERK1/2 MAPK and caspases signaling in cementoblasts. Here, using the flow cytometry assay, we demonstrate that CNTF, acting in a time-dependent manner, influences the apoptosis of cementoblasts: at short-term the survival of OCCM-30 cells was promoted, whereas with long-term CNTF stimulation the apoptotic ratio of cementoblasts increased. The different regulation of CNTF resulting in apoptosis change was due to an activation of mitogen-activated protein kinase ERK1/2 pathway.

Besides its expression in neuronal cells [17], CNTF was also found to be expressed in non-neuronal cells such as osteoclasts, osteoblasts and osteocytes [18]. The presence of the CNTF-receptor complex enables cementoblasts to respond to CNTF, suggesting a role for CNTF in cementum homeostasis through a direct action on cementoblasts. This finding highlights the importance of the presence of CNTFR on cementoblasts, indicating a pleiotropic role of this cytokine in cementum repair. Consistent with previous reports, CNTF induced CNTFR expression in cultured osteoblast-like cells [18]. LIFR was detected in osteoblast-lineage cells in murine osteoblasts [19] and rat pre-osteoblasts [20]. Upon binding to CNTF, CNTFRα triggers the formation of a heterodimer between LIFR complex and GP130 [21], which activates downstream signaling molecules. Our results are the first data to demonstrate that OCCM-30 cells express at mRNA and protein levels for CNTF-receptor complex including CNTFRα, GP130, LIFR as well as IL-6Rα. According to the IF staining results, we presume that CNTF functions by binding with high affinity to the receptor complex (CNTFRα, GP130 and LIFR) and with low affinity to IL-6Rα.

The functional receptor complexes for IL-6 cytokines share a membrane GP130 as a critical component for signal transduction [22]. In CNTF-receptor complex, GP130 and ligand-specific chains are related to cytoplasmic tyrosine kinases [22]. In the present study, it was shown that CNTF induced up-regulation of p-GP130, which has expression that is time-dependent like p-ERK expression. We assume that CNTF stimulation as well as CNTF-receptor complex activation trigger homo- or heterodimerization of GP130, activating the associated cytoplasmic tyrosine kinases with subsequent modification of transcription factors [22].

Many trials reported that exogenous CNTF participates in bone metabolism [23,24]. CNTF mildly inhibited osteoblast differentiation, suppressed osteoblast transcription factor *Osterix* expression and reduced the mineralization formation [23]. Human dental pulp stem cells (DPSCs), mesenchymal cells capable of cementoblastic differentiation potential, have been shown to secrete significantly higher levels of neurotrophic factors [25]. Our results indicated that CNTF impaired the OCCM-30 cell viability and proliferation and enhanced the apoptosis, suggesting that prolonged CNTF stimulation impairs cementoblast homeostasis. During the first stage of CNTF stimulation until 8 h, the *Caspase-8* gene expression was strongly up-regulated. In the second phase (16–32 h), the up-regulation of *caspase-8* increased moderately and exhibited characteristics of a timer-strategy. We could also observe that *caspase-9* was significantly increased in the second phase. These data indicate that CNTF induces apoptosis through caspase-8 in the short-term stage, but relies more on caspase-9 in the long-term stage.

Furthermore, we found that CNTF stimulated transient p-ERK1/2 expression during short-term exposure period. Meanwhile, the apoptosis ratio is down-regulated by CNTF after short-term stimulation. Interestingly, these results suggest that ERK1/2 plays an important role in apoptosis regulation of cementoblasts. Activation of ERK1/2 has been shown to inhibit apoptosis in response to different stimuli [26]. The mechanism by which ERK1/2 activation inhibits apoptosis is complicated and varies, depending on the cell type and the cellular regulatory influences that the cell receives [26]. ERK1/2 controls cell survival or apoptosis by regulating the activity of anti- and pro-apoptotic transcription factors [27]. For instance, Allen et al. (2003) reported that ERK1/2 inhibits apoptosis by caspase-9 and -3 suppression. Hartel et al. (2010) showed that ERK1/2-mediated inhibition of the pro-apoptotic Caspase-3 protects endothelial cells from apoptosis under hypoxia [28]. Our results support this type of mechanism. First of all, FR180204 enhanced the apoptosis of cementoblasts, indicating that ERK1/2 participates in the apoptosis process in cementoblasts. Secondly, we observed that p-ERK1/2 is stable in the presence of CNTF at short-term period stimulation. In addition, our flow cytometric data show that CNTF impaired the FR180204-induced apoptosis ratio. We thus conclude that during short-term CNTF stimulation, CNTF requires p-ERK participation to maintain cell survival. Our data demonstrated that CNTF activates p-ERK1/2, thus regulating the apoptosis process possibly through a cross-talk between ERK1/2 and caspases signaling.

In the flow cytometric analysis of ERK1/2 inhibitor to cementoblasts, all the experimental groups were stimulated with reagents dissolved in DMSO due to its utility as a solvent. It was reported that DMSO potentiates apoptotic signaling pathways [29] and has an overall inhibitory effect on cell proliferation [30]. Thus, the effect of CNTF on cementoblasts apoptosis would be altered due to DMSO addition. However, the aim of the inhibition experiments is to clarify the involvement of ERK1/2, and we thus focused on the regulatory effects of FR180204 on the apoptotic rate of cementoblasts.

Orthodontic forces induce apoptosis of cementoblasts and in turn conditionally initiate OIIRR. Our study raises the possibility of a short period survival protection role of CNTF in cementoblasts mediated via the ERK1/2 and caspases signaling network. Thus, CNTF could be a factor to protect cementum from OIIRR. In this regard, we propose that CNTF initially targets cementoblasts and subsequently induces ERK1/2 and caspases activation through its receptor complex and signal transducer GP130 to regulate cementoblasts apoptosis.

However, one of the limitations of the present study is the lack of the ERK1/2 knockdown assay that could better clarify the involvement of ERK1/2 signals in the interaction of caspases signaling. Thus, further investigations are required to better understand the potential involvement of these pathways in the regulation of apoptosis of exogenous CNTF on cementoblast homeostasis maintenance. Moreover, the exact mechanism of which CNTF-receptor is involved in the modulation of intracellular pathways activation is unclear. The underlying transcriptional activities of overexpressed CNTF-receptor induced by exogenous CNTF stimulation needs to be further investigated. Deeper studies should be conducted to clarify their specific regulatory role by means of knock down technique for the CNTF-receptor complex.

## 4. Materials and Methods

### 4.1. Cell Culture

An immortalized murine cementoblast (OCCM-30) cell line [31,32] was kindly provided by Prof. M. Somerman (Laboratory of Oral Connective Tissue Biology, NIH, Bethesda, MD, USA). OCCM-30 cells were cultured in α-minimal essential medium (α-MEM) (11095-080, Gibco, Gaithersburg, MA, USA), supplemented with 10% (*v*/*v*) fetal bovine serum (FBS) (10270-106, Gibco) and 1% Penicillin/Streptomycin (P/S) (15140-122, Gibco) in 6-wells cell culture plates (657160, Greiner bio-one, Frickenhausen, Germany) at 37 °C humidified atmosphere with 5% CO_2_. The growth medium was replaced every 2 d with fresh standard medium. Until proliferatory outgrowth of adherently growing cementoblasts was observed, the cells were seeded at a density of 1 × 10^6^ cell/well. The concentration of FBS was reduced to 0.5% as starvation medium 12 h prior to cell stimulation. All experiments were performed using cells at the 2nd to 5th passages.

### 4.2. Cell Stimulation and Pharmacological Inhibitor

OCCM-30 cells were stimulated using different concentrations of recombinant mouse CNTF protein from Novus Biologicals (Cat. N°: NBP2-35168). Purity > 95% (Determined by SDS-PAGE). Endotoxin < 1.0 EU/μg (Determined by the LAL method). Source: *Escherichia coli*. Formulation: Lyophilized from sterile water. This protein was reconstituted following manufacturer indications to a stock solution of 1.0 mg/mL in sterile water and stored at −20 °C. The proinflammatory cytokine mouse recombinant interleukin-6 (IL-6) (Cat. N°: I9646, Sigma-Aldrich, Taufirchen, Germany) was applied as a positive control for cell viability and proliferation assay.

For the ERK1/2 inhibition experiments, ERK inhibitor (FR180204) (#328007, Calbiochem, CA, USA) was used at 1.0 μg/mL and the same amount of DimethyIsulfoxide (DMSO) (D8418-50ML, Sigma-Aldrich) was used as negative-control group.

### 4.3. MTS Viability Assay

The possible effect that exogenous CNTF may exert on the cell viability of OCCM-30 cells was examined calorimetrically using 3-(4,5-dimethylthiazol-2-yl)-5-(3-carboxymethoxyphenyl)-2-(4-sulfophenyl)-2H-tetrazolium (MTS) assay (G3582, Promega, Fernwald, Germany) as previously described [33,34]. Briefly, cells were treated with different concentration of CNTF (0, 50, 100, 200, 400 and 800 ng/mL) in starvation medium over a period of 24 h. IL-6 cytokine served as the positive control. Thereafter, 20 μL of the MTS Reagent was added into each well and the cells were incubated during 2 h at 37 °C in a 5% CO_2_ atmosphere. The MTS optical density (OD) at 490 nm was recorded using a 96-well micro-plate reader (BioTek, Winooski, VT, USA).

### 4.4. Cell Proliferation Assay

Cell proliferation was assessed using Ki-67 immunofluorescent staining as previously described [35]. OCCM-30 cells (5 × 10^3^ cells/well) were seeded onto sterile Falco^TM^ Chambered Cell Culture Slides (354108, Fisher Scientific) overnight to allow adherence and then treated with CNTF protein (Cat. N°: NBP2-35168, Novus Biologicals) for 24 h in starvation medium. The cells cultivated in starvation medium alone served as positive control and those in growth medium were used as negative control. The treated cells were then fixed with Fixation Buffer (554655, BD Cytofix^TM^, Thermo Fisher Scientific, Leipzig, Germany) for 15 min at room temperature (RT) and washed twice with 1 × phosphate-buffer saline (PBS) with 0.02% Tween-20 (P1379, Sigma-Aldrich). Triton^TM^ X-100 (T-9284, Sigma-Aldrich) was added at 0.2% (*v*/*v*, in PBS) and used for permeabilization. Cells were blocked with immunofluorescence blocking buffer (#12411, Cell Signaling Technology, Frankfurt a. Main, Germany) for 20 min at RT. Afterwards, rabbit anti-Ki-67 antibody (ab70362, Cell Signaling Technology) was added to cells at a dilution of 1:50 and incubated overnight at 4 °C. OCCM-30 cells were further stained with DyLight^@^ 488 polyclonal goat anti-rabbit IgG H&L (ab96899, Abcam, Cambridge, UK) at a dilution of 1:500 for 1 h in the dark and were exposed for 5 min to 0.1% Medium with 4′,6-diamidino-2-phenylindole (DAPI) (ab104139, Abcam) for blue nuclear staining. Slides were imaged with confocal high-resolution fluorescence microscopy (Leica Microsystems, Wetzlar, Germany) and positively stained cells were counted manually.

### 4.5. RNA Isolation, Reverse Transcription and RT-qPCR

Total RNA was isolated from CNTF treated-cells using ReliaPre^TM^ RNA Miniprep Systems (Z6011, Promega), and the RNA concentration in each sample was measured on a NanoDrop 2000 Spectrophotometer (Thermo Fisher Scientific).

For cDNA synthesis, total RNA (1 μg) was then reversed-transcribed to yield cDNA by using iScript^TM^ cDNA Synthesis Kit (1708891, Bio-Rad, Feldkirchen, Germany). Next, every RT-qPCR amplification were performed using CFX96^TM^ Real-Time System (C1000^TM^ Thermal Cycler, Bio-Rad) with SsoAdvanced^TM^ Universal SYBR Green Supermix (1725270, Bio-Rad). Primers were designed by Bio-Rad including *CNTFR* (qMmuCED0024700), *IL6Ra* (qMmuCID0005519), *LIFR* (qMmuCID0005702), *IL6st* (qMmuCED0045761), *Caspase-3* (qMmuCED0047599), *Caspase-8* (qMmuCID0005542) and *Caspase-9* (qMmuCED0046922).

Reference gene *PPIB* (qMmuCED0047854) which has been shown to be the most reliable reference gene for normalization in OCCM-30 was used [36]. The relative gene expression (Rel. mRNA) used for the statistical analysis was normalized to the expression of *PPIB* and determined using the standard curve method (2^−ΔΔCq^).

The detailed protocol of RT-qPCR was as follows: 95 °C/30 s followed by 40 cycles of 95 °C/15 s and 60 °C/30 s. The melting curves after each RT-qPCR cycles that were carried out between 50 °C and 95 °C with a plate read every 0.5 °C increment after holding the temperature for 5 s with continuous fluorescence acquisition.

### 4.6. Immunofluorescence (IF)

CNTF treated-cells were fixed with fixation buffer (554655, BD Cytofix^TM^) for 10 min and permeabilized with 0.5% Triton™ X-100 Surfact-Amps™ Detergent Solution (28313, Thermo Fisher Scientific) for 20 min at RT. Then, cells were kept in blocking buffer (#12411, Cell Signaling Technology) for 30 min at RT and washed twice with 1 × PBS with 0.02% Tween-20 (PBST) (P1379, Sigma-Aldrich) for each step. Afterwards, cells were incubated with anti-CNTF Receptor alpha (ABIN3016491, antibodies-online GmbH, Aachen, Germany), LIFR (#50423-T48, Sino Biological, Eschborn, Germany), IL-6R alpha/CD126 (#39837, Cell Signaling Technology), GP130 (#3732, Cell Signaling Technology), Phospho-GP130 (Ser782) (PA5-64501, Invitrogen, Leipzig, Germany) at dilutions of 1:250 at 4 °C overnight. After washing three times with 1 × PBST for 5 min, the cells were incubated with DyLight^@^ 488 goat anti-rabbit polyclonal secondary antibody (ab96899, Abcam) at a dilution of 1:500 which conjugated to fluorescein isothiocyanate for 1 h in the dark at RT. DNA was stained using DAPI (ab104139, Abcam). Slides were imaged using a high-resolution fluorescence microscope (Leica Microsystems, Wetzlar, Germany) and photographed.

### 4.7. Apoptosis Detection with Flow Cytometry Analysis

Apoptotic cells after treatment with CNTF were detected and quantified using the reagent Propidium Iodide (PI) from the Annexin-V FITC/PI Apoptosis Detection Kit (#556547, BD Biosciences, Heidelberg, Germany) as described previously [14,37]. The reagent PI is used for staining dead cells with red fluorescence, as PI binds tightly to the nucleic acids, which are not accessible in living cells. In short, both floating and adherent cells were harvested with Gibco™ StemPro Accutase Cell Dissociation Reagent (A1110501, Gibco). For staining control and experimental groups, cells were stained with 5 μL Annexin-V FITC and 5 μL PI staining solution. Following incubation for 15 min at RT in the dark, 400 μL of 1 × binding buffer was added to each tube. Finally, cells were analyzed for apoptosis using a FACS Vantage Flow Cytometer (SP6800 Spectral Analyzer, Sony Biotechnology, Berlin, Germany) with 488 nm excitation for PI. The data was analyzed by FlowJo software 1.0 (Tree Star Inc., California, CA, USA).

The percentages of the different cell populations were processed in the different quadrants in an Annexin-V FITC/PI dot plot using SP6800 Spectral Analyzer software 2.0.2 (Sony Corporation, Tokyo, Japan). For every sample, 1 × 10^4^ events were recorded and this assay was done in triplicate. Cells that were Annexin-V FITC^-^/PI^-^ were considered being alive and cells being Annexin-V FITC^+^/PI^+^ were necrotic.

### 4.8. Western Blotting (WB)

OCCM-30 cells were lysed in Pierce^TM^ RIPA buffer (89901, Thermo Fisher Scientific) supplemented with 3% phosphatase and protease inhibitors cocktail (78442, Thermo Fisher Scientific). The protein concentrations in the lysates were measured quantitatively using Pierce^TM^ BCA Protein Assay Kit (23225, Thermo Fisher Scientific) on a Nanodrop 2000 Spectrophotometer. The protein lysates (20 μg/lane) were then separated by electrophoresis via sodium dodecyl sulphate-polyacrylamide gel (SDS-PAGE) on 10% (*w*/*v*) on 4–20% Mini-PROTEAN^@^ TGX^TM^ Precast Gel (#4561093, Bio-Rad), followed by transfer onto nitrocellulose membranes (1704271, Bio-Rad).

The blotted membranes were blocked with Tris-buffered saline containing 5% (*w*/*v*) non-fat milk (T145.1, ROTH, Karlsruhe, Germany) with 0.05% (*v*/*v*) Tween-20 (P1379, Merck, Darmstadt, Germany) and incubated for 1 h at RT. Specific antigens were immunodetected by the following antibodies: anti-CNTF Receptor alpha (ABIN3016491, antibodies-online GmbH), LIFR (#50423-T48, Sino Biological), IL-6R alpha/CD126 (#39837, Cell Signaling Technology), GP130 (#3732, Cell Signaling Technology), Phospho-GP130 (Ser782) (PA5-64501, Invitrogen), Phospho-p44/42 MAPK (ERK1/2) (Thr202/Tyr204) (#4370, Cell Signaling Technology), p44/42 MAPK (ERK1/2) (#4695, Cell Signaling Technology), Caspase-3 (#9662, Cell Signaling Technology), Caspase-8 (#4790, Cell Signaling Technology), cleaved-caspase 8 (#8592, Cell Signaling Technology) and Caspase-9 (#9504, Cell Signaling Technology) at dilutions of 1:1000. The standardized protein loading was controlled by β-actin (ab8227, Abcam) at a dilution of 1:2000. Polyclonal Goat Anti-Rabbit (P0448, Dako, Santa Clara, CA, USA) HRP was used as the secondary antibody at a dilution 1:2000.

Visualization of the protein signals were conducted with chemiluminescence utilizing Amersham ECL Western Blotting Detection Reagents (9838243, GE Healthcare Life Sciences, Freiburg, Germany) and exposure to Amersham Hyperfilm (28906836, GE Healthcare) on OPTIMAX X-Ray Film Processor (11701-9806-3716, PROTEC GmbH, Oberstenfeld, Germany). Finally, ImageJ software 2.0 (National Institutes of Health, Washington, DC, USA) was used to analyze the intensities of the labelled protein bands.

### 4.9. Statistical Analysis

For statistic evaluation, GraphPad Prism 8.0 software (GraphPad Software Inc., San Diego, CA, USA) was used. Quantitative data are presented as means ± standard deviation (SD) and analyzed using independent and paired Student’s *t*-tests to compare the differences between results in the CNTF-treated and the control groups. A significance level of *p* value of <0.05 was considered statistically significant. Each experiment was performed in triplicate and reproduced at least twice.

## 5. Conclusions

Together, the data presented in this study demonstrates that CNTF-receptor complexes (CNTFRα, GP130, LIFR) are expressed in cementoblasts. Exogenous CNTF targets to cementoblasts to trigger a cascade of ERK1/2 and caspases signaling events leading to cell survival and apoptosis. The deeper mechanism of how CNTF stimulates the signaling cascades through its interaction with its over-expressed receptor complex should be elucidated. Moreover, investigations focusing on the downstream signaling event induced by CNTF and the mechanisms involved in its detrimental impacts on cellular physiology will enhance our knowledge to develop preventive strategies or therapies for OIIRR.

## Figures and Tables

**Figure 1 ijms-23-08335-f001:**
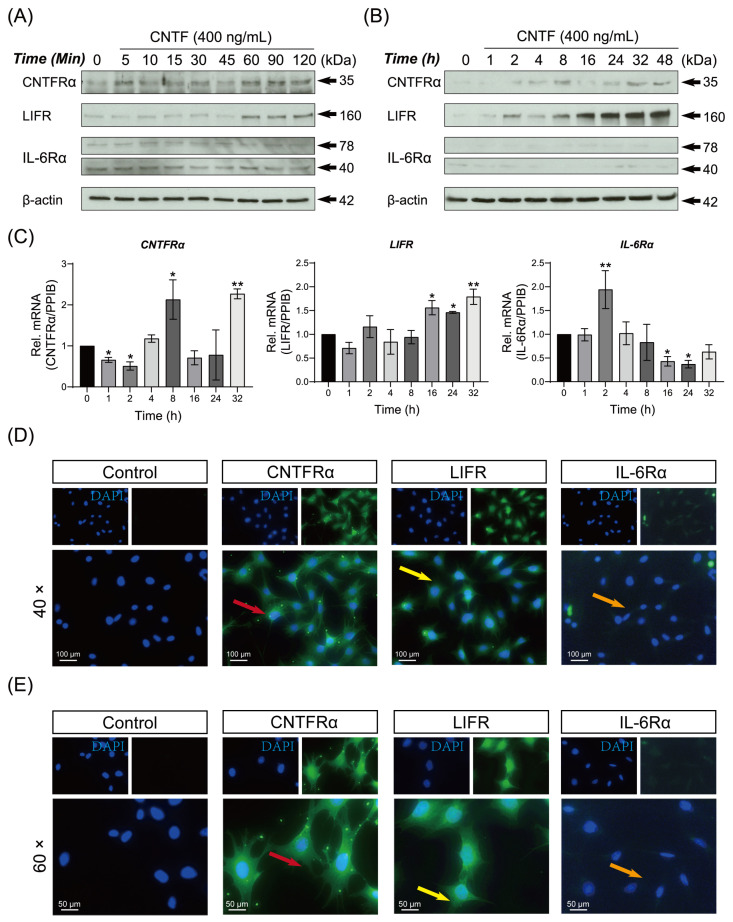
The tripartite CNTF-receptor complex is up-regulated by ciliary neurotrophic factor in cementoblasts. (**A**,**B**) WB showed protein expression of the CNTF-receptors (CNTFRα, LIFR and IL-6Rα) in OCCM-30 cells induced by CNTF protein (400 ng/mL) for various periods. Internal β-actin serves as loading control. (**C**) The expression of mRNAs encoding the CNTF-receptors were quantified by RT-qPCR. The relative mRNA expression of each gene was obtained through normalizing to internal *PPIB*. The statistical significance was determined by student *t*-test (*n* = 3 for each group). (**D**,**E**) The CNTF-receptors immunofluorescent localization showed the expression of CNTFRα (red arrow), LIFR (yellow arrow) and IL-6Rα (orange arrow) in CNTF-treated OCCM-30 cells. Nuclei are stained with DAPI (blue). Scale bar: 100 μm (image magnification: 40×); 50 μm (image magnification: 60×). Bar indicates values ± standard deviation (SD) which represent three independent experiments. Statistically significant differences (indicated by asterisks) are shown as follows (* *p* < 0.05; ** *p* < 0.005).

**Figure 2 ijms-23-08335-f002:**
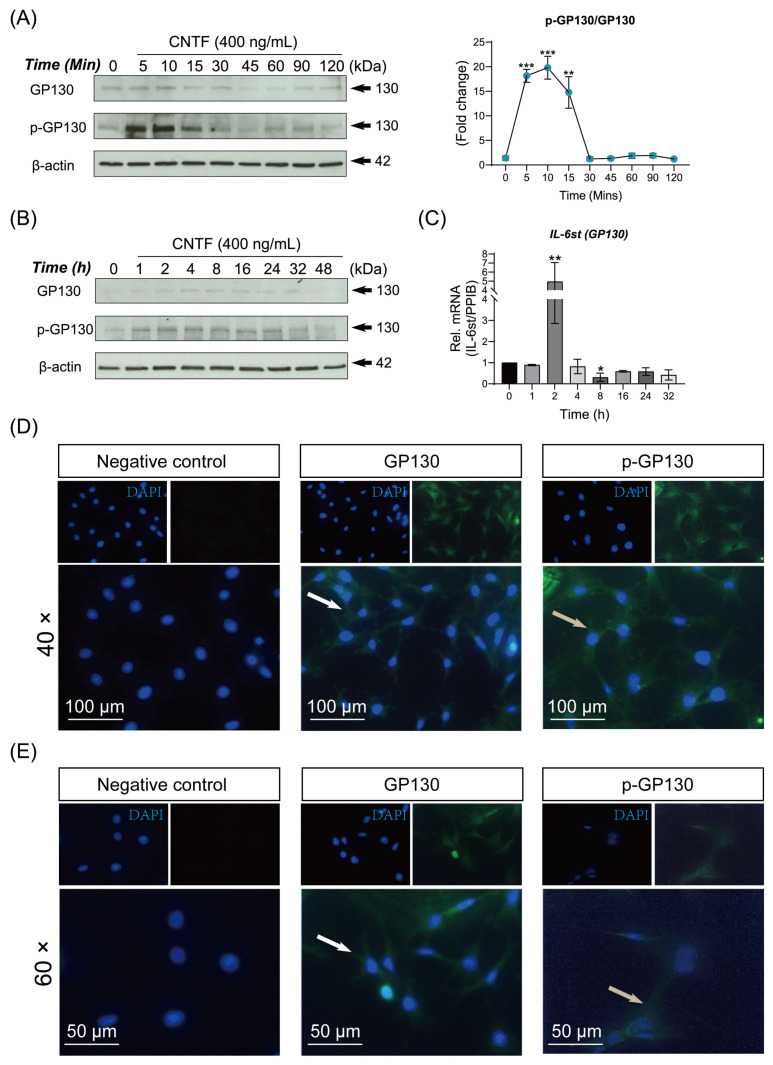
Ciliary neurotrophic factor triggers GP130 protein expression and phosphorylated GP130 in cementoblasts. (**A**,**B**) The protein expression of GP130 and phosphorylated GP130 were determined by WB. Internal β-actin serve as loading control. The line chart shows the densitometric analysis of p-GP130 expression related to total GP-130 expression. (**C**) RT-qPCR quantification of *GP130* (*IL-6st*) gene expression in OCCM-30 cells when treated with CNTF (400 ng/mL) for indicated time. The relative mRNA expression was obtained through normalizing to internal *PPIB*. (**D**,**E**) IF staining of subcellular localization of GP130 (white arrow) as well as p-GP130 (grey arrow) in non-stimulated cells (negative control) and CNTF-stimulated OCCM-30 cells. Nuclei are stained with DAPI (blue). Individual and merged images of GP130 and p-GP130 are shown. Scale bar: 100 μm (image magnification: 40×); 50 μm (image magnification: 60×). Bar indicates values ± standard deviation (SD) which represent three independent experiments. Statistically significant differences (indicated by asterisks) are shown as follows (* *p* < 0.05; ** *p* < 0.005; *** *p* < 0.0005).

**Figure 3 ijms-23-08335-f003:**
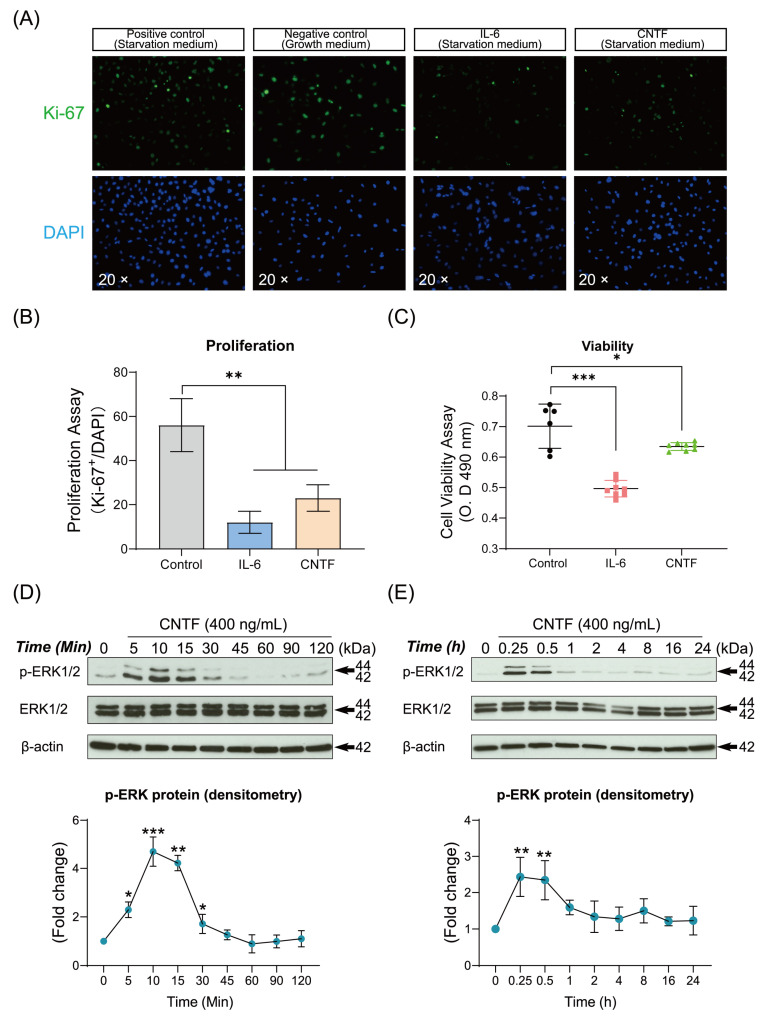
Ciliary neurotrophic factor impairs OCCM-30 homeostasis and activates the expression of ERK1/2 MAPK signaling. (**A**,**B**) Immunofluorescence microscopy images show representative proliferation markers Ki-67 expression. OCCM-30 cells were exposed to CNTF (400 ng/mL) and then underwent immunofluorescence staining for Ki-67 to visualize cells in the proliferation stage. The proportion of proliferating cells for each group was quantified according to Ki-67 positive cells (Ki-67^+^)/total cell counting (DAPI). (**C**) Cell viability assay was performed by MTS assay. IL-6 cytokine served as positive control. (**D**,**E**) Representative immunoblot of p-ERK1/2 protein expression in the presence of CNTF (400 ng/mL) at different time points. Internal β-actin serves as loading control. Densitometric immunoblot analysis of bands indicated the enhanced p-ERK1/2 expression relative to that of the control group. Densitometric results are showed as fold change. Bar indicates values ± standard deviation (SD) which represent three independent experiments. Statistically significant differences (indicated by asterisks) are shown as follows (* *p* < 0.05; ** *p* < 0.005; *** *p* < 0.0005).

**Figure 4 ijms-23-08335-f004:**
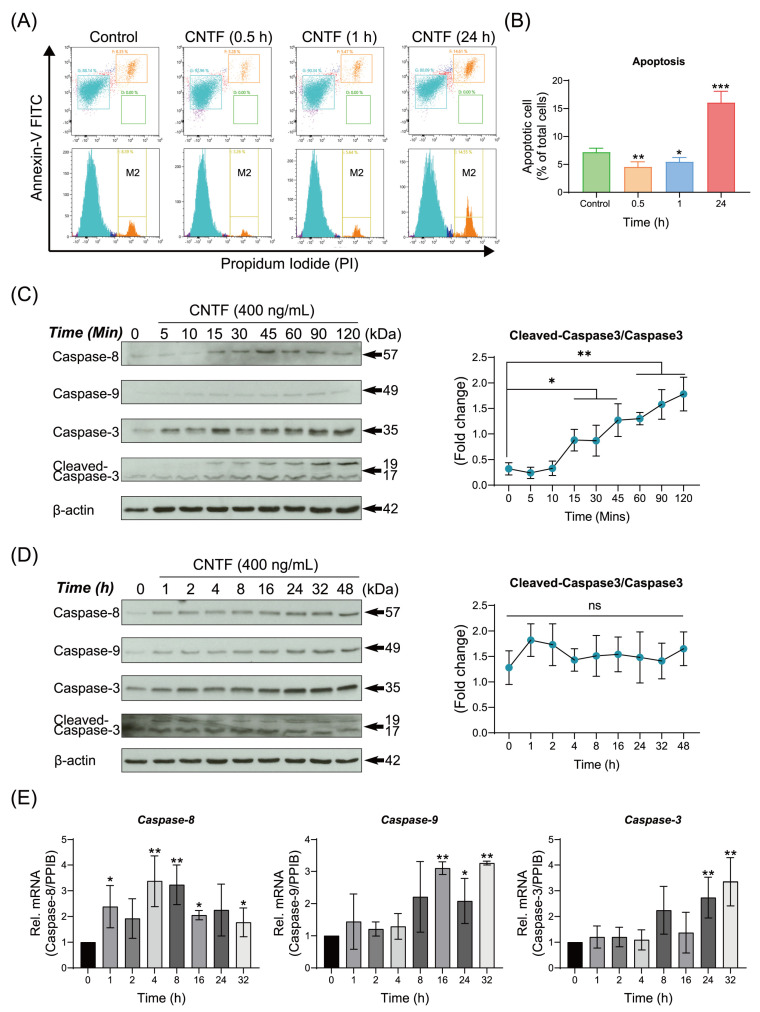
Ciliary neurotrophic factor regulates apoptosis rate and triggers the caspases signaling. (**A**,**B**) Representative plots from Annexin-V FITC and PI staining by flow cytometry analysis performed in triplicate are shown. Apoptotic cells (Annexin-V FITC^+^/PI^+^) are shown in the upper right quadrant. Graphics show the percentages of apoptotic cells exposed to CNTF (400 ng/mL) at different time periods. (**C**,**D**) Representative immunoblot showed that the protein expression of Caspase-8, -9 and -3 as well as cleaved-caspase-3 in response to CNTF (400 ng/mL) in a time-dependent manner. β-actin was loaded as an internal control. (**E**) mRNA expression of *Caspase-8*, *-9* and *-3* in response to CNTF (400 ng/mL) stimulation at indicated time period. Bar indicates values ± standard deviation (SD) which represent three independent experiments. Statistically significant differences (indicated by asterisks) are shown as follows (ns, no significant difference; * *p* < 0.05; ** *p* < 0.005; *** *p* < 0.0005).

**Figure 5 ijms-23-08335-f005:**
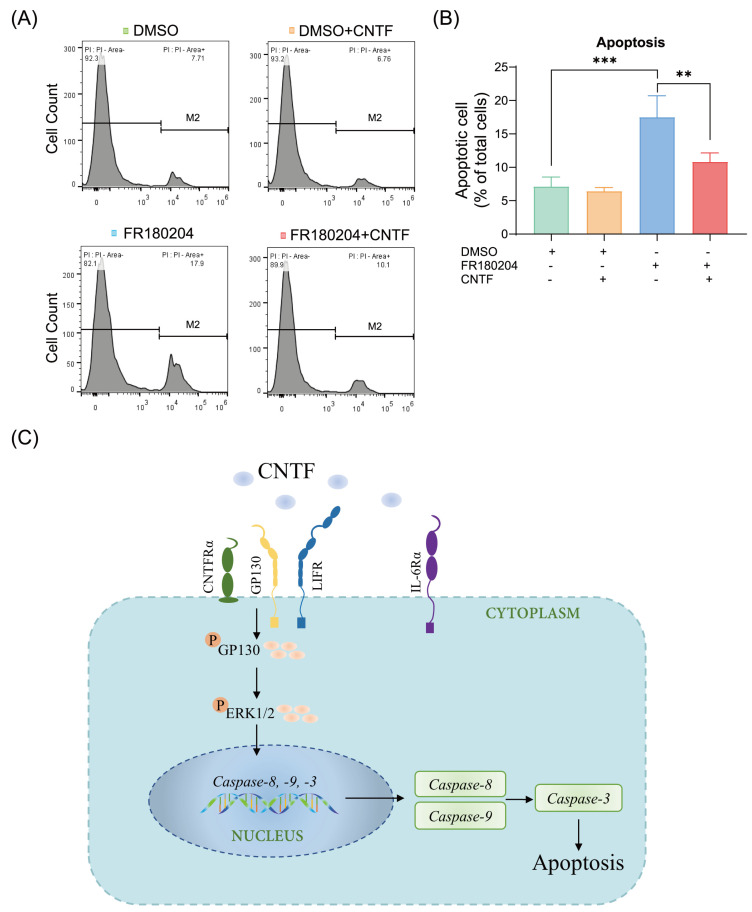
ERK1/2 signal is involved in the regulation of apoptosis of cementoblasts and the caspases pathway. (**A**,**B**) Graphics show the percentages of apoptotic cells exposed to ERK1/2 inhibitor (1.0 μg/mL, FR180204) as well as co-stimulation with CNTF (400 ng/mL). (**C**) The scheme summarizes the mode of CNTF action in cementoblasts: CNTF activated the tripartite CNTF-receptor complex targets and phosphorylated GP130 protein, which recruits ERK1/2 signaling and caspases signaling expression. FR180204 promotes apoptosis in OCCM-30 cells and CNTF addition suppressed the ERK1/2 inhibitor-induced apoptosis within a short-term period. Bar indicates values ± standard deviation (SD) which represent three independent experiments. Statistically significant differences (indicated by asterisks) are shown as follows (** *p* < 0.005; *** *p* < 0.0005).

## Data Availability

The datasets used and/or analyzed during the current study are available from the corresponding author on reasonable request.

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
