# Peer review of "Ciliary Neurotrophic Factor (CNTF) and Its Receptors Signal Regulate Cementoblasts Apoptosis through a Mechanism of ERK1/2 and Caspases Signaling"

_ijms, 2022, doi:10.3390/ijms23158335_

Round 1
Reviewer 1 Report
In the manuscript titled “Ciliary neurotrophic factor (CNTF) and its receptors signal regulate cementoblasts apoptosis through a mechanism of ERK1/2 and caspases signalings ” Yong et al., has used the OCCM-30 cementoblast line to demonstrate that CNTF upregulates apoptosis through caspase and ERK1/2 activation. Currently there are no known receptors for CNTF, hence the mechanisms by which this cytokine signals in cementoblasts remain elusive. Young et al., employs confocal imaging and multiple cell-based assays to elucidate the mechanism of action of CNTF in cementoblasts. While knowledge about pathways operating downstream of CNTF will benefit the researchers in the field, there are several experimental drawbacks in the manuscript. As a result, a major revision is required to ensure that the paper is ready for publication. I have listed the major and minor concerns below:
Major concerns:
1. In lines 225-227, the authors describe that if CFTR signaling trough certain receptors, the expression of the receptors should increase in a time dependent manner upon stimulation with the cytokine. Is it known that CFTR treatment increases the receptor expression in other cells? This is not case for many other receptors. For example insulin treatment of cells does not increase insulin receptor expression.
2. In figures 1(D) and (E) the CFTR and LIFR staining appears to be non specific. Please provide panels after knock-down of the proteins to ensure that the signal observed is a speicifc one. Moreover, the receptors do not have a membrane distribution. LIFR appears to have nuclear distribution. Similar problems are observed in figure 2 (D) and (E). The images look out of focus and staining appears to be like background noise. More controls are needed in the form of siRNA knock-down of the proteins of interest along with secondary antibody staining alone
3. In figure 3(A) why is the negative control showing Ki-67 staining?
4. In figure 4(E) why are the changes in caspase8 expression biphasic?
5. In Figure 4(B) a distinct rise in apoptosis is overserved upon CNTF treatment. Why is is that trend not observed in 5(B)? The orange bar is almost the same height as the DMSO control
6. The finally conclude that CNTF binds to receptors to activate ERK signaling to being about apoptosis. However, further experiments are required to prove that CNTF binding to receptor is bring about this phenomenon. Please show experiments with CFTR/LIFR knock out or knockdown
Minor concerns:
1. There are several typographical errors throughout the manuscript
Reviewer 2 Report
In this manuscript authors evaluated if cementoblasts express CNTF-receptors and analyze the mechanism of its apoptotic regulation effects on cementoblasts. They found that CNTF-receptor complex (CNTFRα, LIFR, GP130) was functionally up-regulated in cementoblasts exposed to exogenous CNTF and CNTF significantly attenuated cell viability and proliferation for long-term stimulation. Moreover, CNTF enhanced the apoptosis after long treatment suggesting a potential role of CNTF as inhibitor of cementoblast homeostasis.
This is a very interesting and well written study but it presents some flaws that must be resolved. In particular:
Introduction: Authors should stress the fact that CNTF and its receptor are also expressed in non-neuronal cells and tissues such as adipocytes (PMID: 31781039), myocytes (PMID: 19136654), prostatic cells (PMID: 33131268) and pancreatic β-cells (PMID: 19272793). This is an important point to introduce because CNTF and CNTFR expressions are mainly linked to neuronal cells. This would further highlight the importance of the results found by the authors showing a pleiotropic role of this cytokine.
Figure 1A and B: Why authors used 400 ng/ml? It is a very high concentration, did they try lower concentrations of CNTF with a dose-response curve? Densitometric analysis must be shown.
Figure 1D and E: negative controls must be shown
Figure 2A and B: Densitometric analysis of pGP130/GP130 must be shown.
Figure 2D and E: Improve image quality. Negative controls must be shown
Figure 4C and D: Densitometric analysis must be shown. In order to prove the apoptosis induction by CNTF, Cleaved Caspase-3 must be shown since it is the cleaved form of this caspase that is active.
Round 2
Reviewer 1 Report
While the authors have introduced some changes to the manuscript, the overall quality of the manuscript still remains unsatisfactory for publication. None of the requested experiments have been performed, which are necessary to support many of the claims that the authors are making.
Major points:
1. Response to point 1 raised in the 1st round of review is not convincing. Are there other papers showing that CNTF treatment increases receptor expression on cells? In most cases treatment with a ligand leads to drop in receptor expression, as activated receptors are removed from the membrane by endocytosis, as a means to switch off signaling. Can the authors explain why CNTF treatment increases receptor expression?
2. The IF panels are still not convincingly showing membrane expression of receptors.
3. Response to point 3 raised in 1st review remain unsatisfactory. Negative control for experiments are done to show a scenario where we should not observe any signal or significantly reduced signal compared to the test samples. However, very distinct Ki67 staining is being observed in the negative control panel (figure 3A). In fact the negative control staining is even better than the IL6 and CNTF treated cells. That begs the question if the negative control experiments have been designed correctly
Reviewer 2 Report
The manuscript has been significantly impoved and can be accepted in the present form.
Author Response
Again, we would like to thank the reviewer 2 for the improvement in the manuscript quality.
Round 3
Reviewer 1 Report
The authors have addressed my concerns and the manusctipt is ready for publication